# Intraosseous Regional Administration of Prophylactic Antibiotics in Total Knee Arthroplasty

**DOI:** 10.3390/antibiotics11050634

**Published:** 2022-05-09

**Authors:** Zoe Wells, Mark Zhu, Simon W. Young

**Affiliations:** 1Department of Orthopedics, North Shore Hospital WDHB, Auckland 0620, New Zealand; mark.zhu@waitematadhb.govt.nz; 2Department of Surgery, University of Auckland, Auckland 1023, New Zealand

**Keywords:** total knee arthroplasty, prophylaxis, intraosseous, regional, antibiotics, IORA, vancomycin, cephazolin, prosthetic joint infection

## Abstract

Prosthetic joint infection (PJI) after total knee arthroplasty is a devastating complication. With the development of antibiotic resistance, a safe and effective means of delivering antibiotic prophylaxis is needed. Intraosseous regional antibiotics (IORA) achieve higher local concentrations of antibiotics with fewer systemic side effects. Previous studies have proven the safety of IORA, whereas animal models have proven it to be more effective than intravenous antibiotics for preventing infection following surgery. Recently, large-scale retrospective studies have also demonstrated lower PJI rates following TKA when IORA is used when compared to routine intravenous prophylaxis. In this article, we propose an evidence-based method for the administration of intraosseous antibiotics in TKA, cover the up-to-date data supporting its use, and explore future directions for additional research.

## 1. Introduction

Total knee arthroplasty (TKA) is one of the most successful surgical inventions of the 20th Century [1]. TKA is the gold-standard treatment for end-stage knee osteoarthritis, with over a million surgeries performed in the USA in 2020 [1]. Advancements in biomaterial technology and biomechanical theory have led to significant improvements in both patient-reported outcomes, and implant longevity. An area of ongoing research is the prevention and treatment of prosthetic joint infection (PJI), a dreaded and devastating complication for patient and surgeon. Despite the widespread use of prophylactic antibiotics and modern sterile techniques, the incidence of PJI remains between 0.8% and 2.5% [2] and appears to be increasing [3]. Intravenous antibiotics are widely used as prophylaxis, but an evaluation of their effectiveness in the last decade demonstrated that tissue concentrations of systemically administered antibiotics do not always meet the minimum requirements for effective prophylaxis [4].

Regional antibiotics for prophylaxis in TKA aim to address this problem by increasing local tissue concentrations at the target site. IVRA, or intravenous regional antibiotics, refers to the administration of antibiotics into the limb of choice beneath an inflated tourniquet. Although its use has been studied in the setting of TKA, it is time-consuming to administer [5,6,7]. The use of intraosseous regional antibiotics (IORA) was pioneered by Young et al. in 2012 [8]. This technique involves delivering antibiotics into trabecular bone beneath an inflated tourniquet prior to skin incision. The safety profile and superior tissue concentrations achieved by this technique are well-proven [8,9,10,11,12,13,14,15]. Recently, large retrospective patient trials have also demonstrated its efficacy for reducing the incidence of PJIs in TKA [16,17].

## 2. Antibiotic Prophylaxis in Arthroplasty

By the 1980s, prophylactic antibiotic use for arthroplasty was known to be effective in reducing PJI [18,19]. The specific regime of systemic intravenous prophylaxis combined with antibiotic cement remains standard protocol worldwide [20]. The causative organisms in PJI are most commonly coagulase-negative *Staphylococcus (CoNS)* and *Staphylococcus aureus (S.a),* which together account for two thirds of PJI worldwide [21]. In combating these organisms, cephalosporins (for example, cefazolin) have been used as prophylactic agents to good effect over the last three decades (reducing PJI from 7.6% to 0.7–0.9% when compared to placebos) [18,22,23]. The goal posts are shifting, however, as increasing antibiotic resistance forces clinicians to re-evaluate their use.

For effective antibiotic prophylaxis, tissue concentrations of the agent must be present at or above the minimum inhibitory concentration (MIC) from the time of skin incision to the time of closure, when contamination is occurring [8,24]. MICs are specific to each pathogen, and as organisms develop antibiotic resistance, their MIC increases. The MIC_90_ (MIC required to prevent the growth of 90% of organisms) of cephazolin for CoNS has previously been reported as 0.5–1.0 µg/mL, but with increased resistance it can be up to 100 µg/mL [4]. With concerns that cephalosporins will not protect patients from PJI caused by MRSA, attention turned to alternative agents such as vancomycin, to which MRSA remains largely sensitive, and for which the MIC for CoNS is only around 2.0 µg/mL [21,25]. Vancomycin has different pharmacological properties to those of cephalosporins. It exhibits concentration-dependent killing, whereas cephalosporins are time-dependent in their action. Put simply, the effectiveness of cephalosporins is determined by how long the serum concentration remains above the MIC at the target site. Further increasing the concentration does not increase effectiveness. Vancomycin, being a concentration-dependent agent, is most effective when the peak concentration is ≥10 times the MIC at the target site. The response of an organism to a concentration-dependent agent can be predicted by the peak concentration:MIC ratio [26,27].

However, vancomycin use presents other challenges. The timing of administration is important; and vancomycin must be administered slowly to avoid reactions such as red man syndrome. In practice, the administration of the standard 1 g prophylactic dose must be initiated at least 1 h prior to the beginning of surgery. This is logistically challenging and leaves the patent inadequately protected when the administration is not timed precisely [12,28]. Garey et al. examined the timing of vancomycin prophylaxis in cardiac surgery and found that only 8% of patients received their dose at the optimum time [29]. Additionally, the use of vancomycin prophylaxis carries the risk of increased incidence and severity of AKI when compared to cephazolin [22,30].

## 3. Regional Antibiotic Delivery

In investigating the ramifications of increasing MICs for cephazolin and Cons/MSSA, Yamada et al. found that when a standard prophylactic dose of 2 g cephazolin was administered IV, bone concentrations during knee arthroplasty did not always exceed the MIC [4]. Tissue samples from the distal femur and proximal tibia exhibited concentrations of cephazolin that met the MIC_90_ for MSSA but not for CoNS (when MIC_90_ was defined epidemiologically from the population studied). Regional antibiotic delivery can achieve higher tissue concentrations with the same total doses of antibiotics. Work has been carried out to specifically examine regional antibiotic prophylaxis in TKA.

The first method of regional antibiotic delivery was via intravenous administration of antibiotics beneath an inflated tourniquet or “Bier’s block” (intravenous regional antibiotic or IVRA). This technique has been used with success to treat osteomyelitis of the extremities in horses and other animals [31]. In 1990, Hoddinott et al. demonstrated that in the context of TKA, administering a cephalosporin antibiotic via a foot vein beneath an inflated tourniquet achieved target tissue concentrations 13 times higher than standard IV prophylaxis [5,32]. Teicoplanin (a glycopeptide antibiotic used across Europe) was studied in the setting of IVRA for TKA in the late 1990s and early 2000s. The MIC of teicoplanin for CoNS is 2–4 mg/L, so it is comparable to the use of vancomycin [6]. De Lalla et al. measured drμg concentrations in skin, fat, bone, and synovium during TKA, and found that even when the dose for systemic prophylaxis was twice that for regional prophylaxis, the tissue concentrations for IVRA were 2 to 10 times greater [6]. They also noted that the MIC for CoNS was not always met in patients receiving systemic prophylaxis. The same group went on to perform an open clinical study to examine the safety and efficacy of their protocols for IVRA and found that all 160 patients who received IVRA prophylaxis for their TKA procedure avoided local or systemic adverse effects. A single patient developed a superficial infection, and no deep infections were observed, with a mean follow-up time of 2 years. They concluded that IVRA prophylaxis for TKA is both safe and effective [7]. A retrospective tissue penetration study further expanded on de Lalla’s 1993 study and compared the regional administration of 200 mg teicoplanin with a systemic dose that was four times higher. They found that tissue concentrations throughout a TKA procedure were up to two times higher in the regional group [33].

Although IVRA has been shown to provide tissue concentrations which more reliably meet the MIC required for prophylaxis in TKA, it does present some procedural challenges. IVRA requires the placement of a foot-vein cannula, which takes additional anaesthetic time and may compromise sterility. In addition, it carries with it the risks of cannulation-related complications such as thrombophlebitis, extravasation and hematoma [27].

## 4. Intraosseous Antibiotic Delivery

Intraosseous cannulation is a technically straightforward and reliable method of delivering fluids and drugs. These attributes have made it a popular technique in trauma and resuscitation scenarios [34]. Its use as a means of delivering local antibiotics as a prophylactic measure was pioneered by Young et al. in 2012 [8]. It was theorized that IORA could address some of the challenges of IVRA and provide superior prophylaxis in TKA. The initial technique proposed by Young’s group used the proximal tibia as the IO injection site. They trialed 1 g of cephazolin in 200 mL of normal saline, injected after tourniquet inflation but immediately before skin incision. This randomized controlled trial compared this IORA group to a systemic-only group, who received 1 g of IV cephazolin prophylaxis at the beginning of the case. Tissue samples were taken from fat and bone at four timepoints throughout the surgery and analyzed. The authors found that the mean fat concentration of cephazolin was almost 17 times higher in the IORA group, and the mean bone concentration was almost 12 times higher [8]. In agreement with previous work performed on IVRA, they found that the tissue concentrations in the systemic group fell well below the MIC_90_ required to inhibit CoNS colonization, whereas the IORA cephazolin concentration did reliably reach the threshold.

The same group continued this line of study to examine delivering vancomycin with the IORA method. They hypothesized that better action against CoNS could be achieved by taking advantage of the lower MIC of vancomycin, while avoiding the prolonged systemic IV administration time typically required [9]. The adverse effects of systemic vancomycin (AKI, neutropenia, red man syndrome) are known to be minimized by reducing the dose and prolonging the infusion time [35]. The authors of that study compared a standard IV dose of 1 g vancomycin with IORA doses of 500 mg and 250 mg vancomycin. They found that the mean bone concentration in the IO groups was higher than that in the IV group, despite lower dosing. Even the lowest IO dose of 250 mg was significantly higher (mean 16 μg/g in the 250 mg IORA group, 38 μg/g in the 500 mg IORA group, compared to 4 μg/g in the systemic group). With the MIC_90_ for CoNS quoted as 2.0 μg/g [21,25], this is a promising outcome. The development of red man syndrome in one patient receiving systemic dosing prompts the question: does IORA reliably protect patients from vancomycin’s adverse effects? Klasan et al. reviewed the procedures of 631 TKA patients, of whom 331 received IORA 500 mg vancomycin for prophylaxis. Comparing those who received IORA vancomycin with those who only received systemic cephazolin, they found no increased incidence of AKI (5% compared to 3%, not statistically significant) or neutropenia in the IORA vancomycin group [15], nor were there any cases of red man syndrome in the study. This is a significant improvement on a comparable study investigating the adverse effects of IV vancomycin prophylaxis, in which there was a rate of 13% AKI compared to only 8% in a cephazolin group [30]. IORA delivery of vancomycin achieves high tissue concentrations, while keeping the dose low and therefore minimizing systemic side effects [15].

Although it could be reasonably assumed that the higher tissue concentrations obtained with IORA would provide a more effective prophylaxis against PJI, Young et al. sought to investigate this in a live animal model [10]. In a 2015 paper, mice were implanted with a bacteria-laden knee “prosthesis” by way of a k-wire inoculated with a standardized number of *Staphylococcus aureus*. Prophylaxis was provided by IV vancomycin or cephazolin, or IORA vancomycin or cephazolin. Both IORA vancomycin and cephazolin were shown to be more effective than their IV counterparts at the same dose, with fewer colony-forming units measured. Low-dose IORA vancomycin was found to be as effective as high-dose IV vancomycin, which can be explained by the concentration-dependent nature of vancomycin and the high tissue concentrations achievable via IO administration [27].

## 5. High BMI and Revision TKA

Certain groups of TKA patients require special consideration when it comes to providing adequate prophylaxis for surgery. One group consists of patients with increased BMI. To reach required therapeutic concentrations, vancomycin dosing must be weight-adjusted [11]. To maintain safe administration, the infusion times must therefore increase, further complicating the perioperative procedure and heightening the risk of inadequate tissue concentration during surgery. Chin et al. randomized patients with elevated BMI (>35) who underwent TKA to receive weight-based IV vancomycin (15 mg/kg up to a max of 2 g) or a standardized 500 mg IORA. They found that the mean tissue concentrations were higher in the IORA group despite the doses being smaller, with a 5% to 9% increase in bone and fat concentrations [11]. They did not find a significant difference in adverse effects, showing that IORA is both safe and effective in an overweight population.

Another group requiring special consideration is those patients undergoing revision surgery who have indwelling tibial components. These patients have an increased risk of infection due to increased surgical duration and complexity [36]. Challenges are also presented by the need to inject the antibiotic into a tibial plateau that may be obstructed by the tibial component, and the need to let the tourniquet down during a prolonged procedure. In 2017 Young et al. performed a small study of 20 revision cases and found that their ability to gain IO access via the proximal tibia was not impaired (although they did propose the medial malleolus as an alternative site if this was found to be difficult) [12]. They found local tissue concentrations of vancomycin to be 5 to 20 times higher in the IORA group, despite the systemic group having a higher dose. Significantly, this advantage was maintained 1.5 h after tourniquet deflation, with the local tissue concentration still 5.3 times higher in the IORA group [12].

## 6. Reduced Tourniquet Use

The use of a tourniquet to create a regional block is part of the basic framework of regional antibiotic administration. In recent years, however, there has been increased interest in performing TKA with reduced tourniquet time. Young et al.’s 2017 paper on TKA revision shows that superior tissue concentrations of vancomycin can be achieved even in revision surgery when the tourniquet is let down during the case [12]. A recent randomized controlled trial compared IORA to IV prophylaxis in TKA with minimized tourniquet time [14]. One group of TKA patients received IV vancomycin and had the tourniquet inflated for cementation only. The second group received IORA vancomycin beneath a tourniquet that was inflated for 10 min following administration, then let down and reflated for cementation only. The mean vancomycin tissue concentrations were 5 to 15 times higher at all timepoints in the IORA group, with bone concentrations prior to cementation of 21.8 μg/g in IORA, compared to 7.9 μg/g in IV. For surgeons wishing to minimize their tourniquet time, IORA remains an option for antibiotic prophylaxis.

## 7. Current Guidelines

There does not yet exist a standardized guideline for the procedure of IORA in TKA that is endorsed by any governing surgical body. However, clinicians may refer to the template laid out in *Essentials of Cemented Knee Arthroplasty* [1] (pp. 664–666). It is made clear that use of IORA must be an adjunct to systemic prophylaxis, which is often mandated by local hospital guidelines. Suggested contraindications to the procedure are patients with large cortical perforations or proximal tibial osteolysis—conditions that would interrupt the metaphyseal venous sinusoids responsible for intraosseous distribution.

The suggested technique uses a manual IO needle (specifically from Cook Medical, Bloomington, IN, USA), though powered options are also appropriate (e.g., EZ-IO, Teleflex Corp, TX, USA).

### 7.1. Antibiotic Solution

As shown by Young et al. 2014 and 2018, and Klasan et al. 2020, a 500 mg IO dose of Vancomycin is appropriate for reaching adequate tissue concentrations without causing adverse effects, even in the event of tourniquet failure and the antibiotic entering the systemic circulation [9,12,15]. This dose is mixed in the operating room with 10 mL of sterile saline, and then made up to 100 mL with saline (Figure 1) on the sterile table. The dose is prepared for injection in two 50 mL syringes (Figure 2).

### 7.2. Procedure

The standard location for injection is the proximal tibia, though the medial malleolus has been suggested as an alternate site if needed. The operative leg is prepped and draped in the usual fashion for TKA. Exsanguination is performed and the tourniquet is inflated to 250–300 mmHg. A small hole is then made in the sterile adhesive bandage over the medial aspect of the proximal tibia (Figure 3). The IO needle may now be inserted through the skin without dragging or twisting the adhesive bandage. The specific placement of the IO needle should be just medial and proximal to the tibial tubercle (Figure 4). This location allows for easy penetration as there is thinner cortical bone proximally. In revision cases, a slightly more distal insertion point may be necessary. Revision cases may also necessitate the use of a powered IO driver due to thicker cortical bone post-implantation.

Once inserted, the injection solution can be connected to the IO needle and run over the course of 1–2 min (Figure 5). The IO needle may then be removed, and a sterile dressing placed over the site (Figure 6), allowing surgery to proceed as usual.

## 8. In Clinical Practice

In their 2020 study of 631 patients, Klasan et al. found PJI rates for IORA patients to be lower than for IVA patients in a comparative systemic prophylaxis population [15]. They found a PJI rate of 0.2% for IORA at 1 year follow-up, compared with the quoted 1-year PJI rates of 0.74% [37] and 1.2% [38] in large studies. Although this result is promising, the numbers in their study were too small to draw meaningful conclusions on the effectiveness of IORA. Parkinson et al. retrospectively studied a cohort of 1909 TKAs who received surgery across five separate hospitals in Australia [17]. As five surgeons across these centers adopted the practice of IORA, patients became eligible for the IO group chronologically. Specifically, 1181 patients were included in the IV-only group, whereas 725 patients received IORA. Of those, of 232 received only IORA with no IV prophylaxis at the time of surgery [10]. The specific doses used were either 500 mg IO vancomycin, or 1 g IO cephazolin. To specifically assess the success of IORA prophylaxis they defined their primary outcome as PJI developing within 12 months, as infections during this period are most likely secondary to contamination at the time of surgery. At 12 months, the rate of PJI was 1.4% in the IV only group, and only 0.1% in the IORA group (RR 0.01, 95% CI 0.01 to 0.77. *p* = 0.03). It was recognized that any hematogenous infection occurring within 1 year is a potential confounder, but this would affect both the study and control groups in equal part. They recorded no difficulties in administering IO antibiotics to any of their patients, and there were no adverse effects from the procedure.

Park et al. also found a reduction in PJI rates in a retrospective, multi-center review of 1060 patients undergoing TKA [39]. This study followed 488 patients receiving IORA (IO vancomycin combined with an IV cephalosporin), and 572 receiving IV prophylaxis only. At 90 days post-op, the PJI rate was 1.46% in the IV group, compared to 0.22% in the IORA group (RR 0.15, *p* = 0.047). At 12 months, the risk of PJI was again lower in the IORA group; however, this difference was not statistically significant (2.04% in IORA, 0.37% in IV, *p* =0.07). The failure to reach statistical significance at 12 months may be due to loss to follow-up of almost one third of patients in the IV group. It was noted that the case for IORA’s success may be bolstered by the fact that antibiotic cement was used in most IV prophylaxis cases, but in fewer IORA cases. As in the study by Parkinson, the authors observed no adverse effects from IO administration. However, they noted that using the distal femur for injection was technically difficult and towards the end of the study the proximal tibia was used exclusively.

## 9. Discussion

Over the past decade, IORA delivery of antibiotics has been shown to be safe and effective in total knee arthroplasty. Stemming from concerns about increasing antibiotic resistance and inadequate bone penetration, IORA has been proven to provide adequate tissue concentrations of antibiotics to meet the requirements for effective prophylaxis [8,9,10,12]. Although recent retrospective studies have shown PJI rates to be reduced in IORA groups [17,39], large randomized controlled trials are needed to bring IORA into mainstream clinical practice. Both Young and Symonds have acknowledged that the introduction of IORA prophylaxis will likely occur alongside traditional systemic prophylaxis initially, in order to abide by hospital guidelines and avoid ethical dilemmas [1,27].

Questions remain regarding whether it is appropriate to use IORA for all patients, or whether it should be withheld for only those with high-risk profiles. Studies have shown that in cases in which MSSA is the causative organism for infection, vancomycin does not offer better protection than cephazolin alone [10]. Cephazolin is easy to administer via the systemic IV route and does not require extended infusion time, as vancomycin does. The mechanism of action of Cephazolin is time-dependent, not concentration-dependent. Although the initiation of bacterial killing occurs earlier with higher concentrations of cephazolin, increasing tissue concentrations above MIC may not improve efficacy. Only in select cases, in which the MICs of CoNS for Cephazolin are high, will IORA afford better protection than IV.

Concerns about antimicrobial stewardship suggest that using vancomycin in groups which are low-risk for CoNS infection will promote the development of microbial resistance. Due to this concern, Young et al. proposed in 2015 that IORA vancomycin may be best reserved for patients at high risk of PJI. This includes patients with increased BMI, those undergoing revision surgery, or with significant systemic comorbidities [10]. Although IORA has a favorable safety profile for these patients, its effectiveness has not been well studied in this cohort. Parkinson et al.’s recent retrospective study broke down outcomes by high-risk subgroups, such as impaired renal function, increased BMI, and diabetes. They did not find that these groups were afforded superior protection from PJI when compared to a low-risk cohort [11,12,17]. In each case there were fewer infections in the IORA group, but the outcomes failed to reach statistical significance. This research is underpowered due to the much smaller cohorts available in these high-risk groups. In the future, larger studies will be needed in order to draw conclusions on the efficacy of IORA in high-risk groups. In agreement with the previous work carried out by Young’s group [11,12], they did not find increased rates of adverse events in the high-risk groups, suggesting that IORA is still safe for these patients.

IORA has been trialed in the context of revision TKA and was found to be both safe and provisionally effective as a means of prophylaxis [12]. The ability to achieve high tissue concentrations of antibiotics in the context of existing implants indicates that IORA may be useful for treating PJIs as well. In cases in which there are implants in situ, the treatment of infection must take into account the concept of biofilm, that is, microbial cells aggregated irreversibly onto a surface and enclosed by a polysaccharide matrix [40]. To combat PJIs, the antibiotic concentration needs to reach the MBEC (minimum biofilm eradication concentration). This is often not possible with IV antibiotics as the doses required would be systemically toxic. Kildow et al. used IORA as an adjunct in a group of patients undergoing DAIR (debridement, antibiotics, and implant retention) for PJI [41]. They retrospectively reviewed 26 patients with early PJI (within 3 months of initial surgery) and found a 92.3% success rate of DAIR with concurrent IORA therapy (mean follow-up time of 15.3 months). It is difficult to compare this result with those of other studies as the rate of success of DAIR in early PJI has been reported to range from less than 50% to more than 90% [42]. Importantly, however, no adverse effects of IORA were reported. The authors also found administration of IORA to be straightforward, even in the group with complex implants [41]. There is scope, therefore, for further work in this area.

IORA is proven to be a safe and practical means of delivering antibiotics. It achieves significantly higher tissue concentrations in TKA when compared to IV prophylaxis. Although this appears to translate into a reduction in the risk of PJIs, the evidence for this was generated from animal studies and retrospective cohort studies. A large, adequately powered, randomized controlled trial is needed to further validate its use.

## Figures and Tables

**Figure 1 antibiotics-11-00634-f001:**
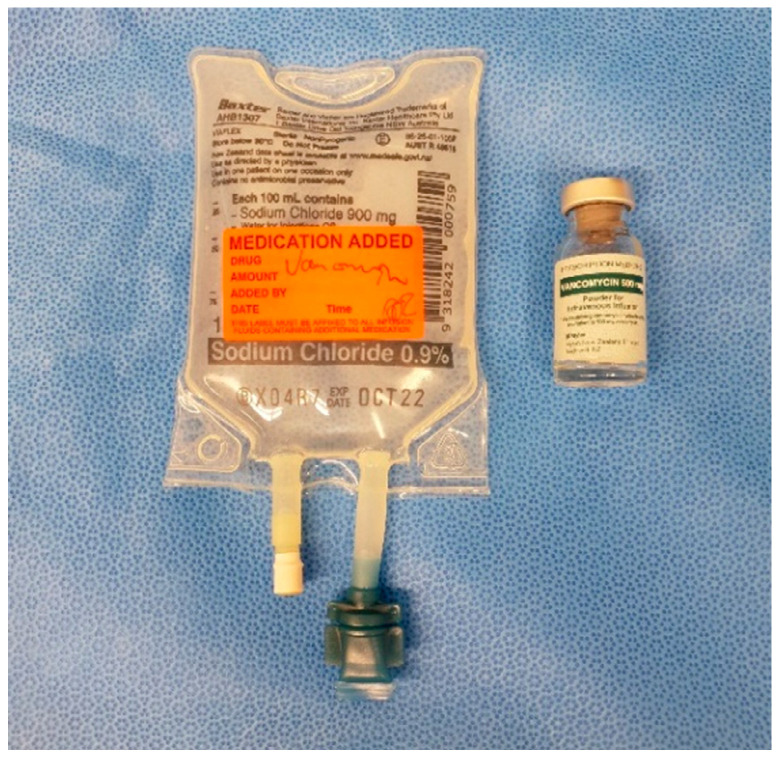
500 mg vancomycin powder in 100 mL saline.

**Figure 2 antibiotics-11-00634-f002:**
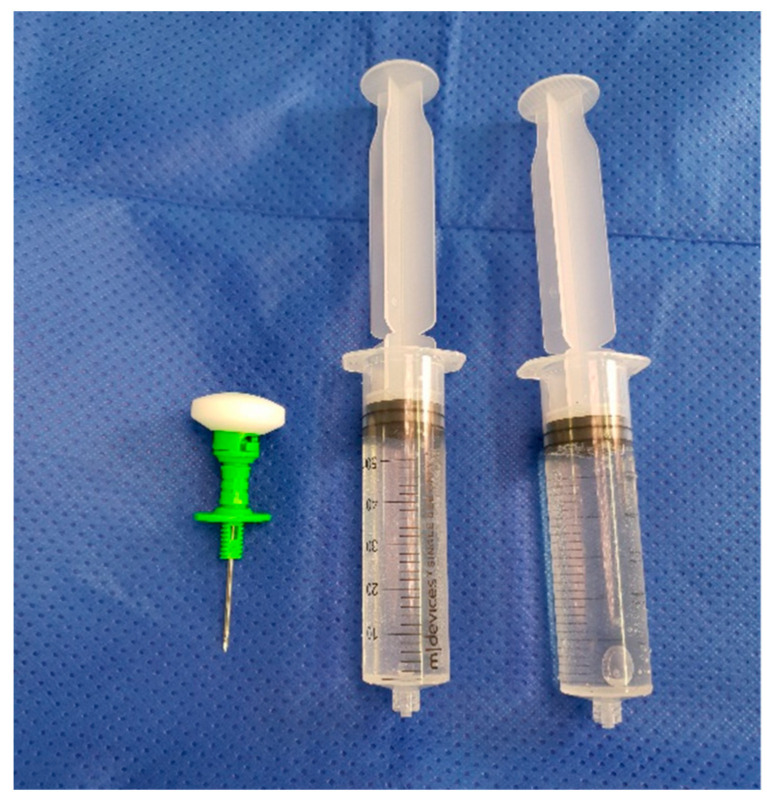
Solution prepared in 50 mL syringes.

**Figure 3 antibiotics-11-00634-f003:**
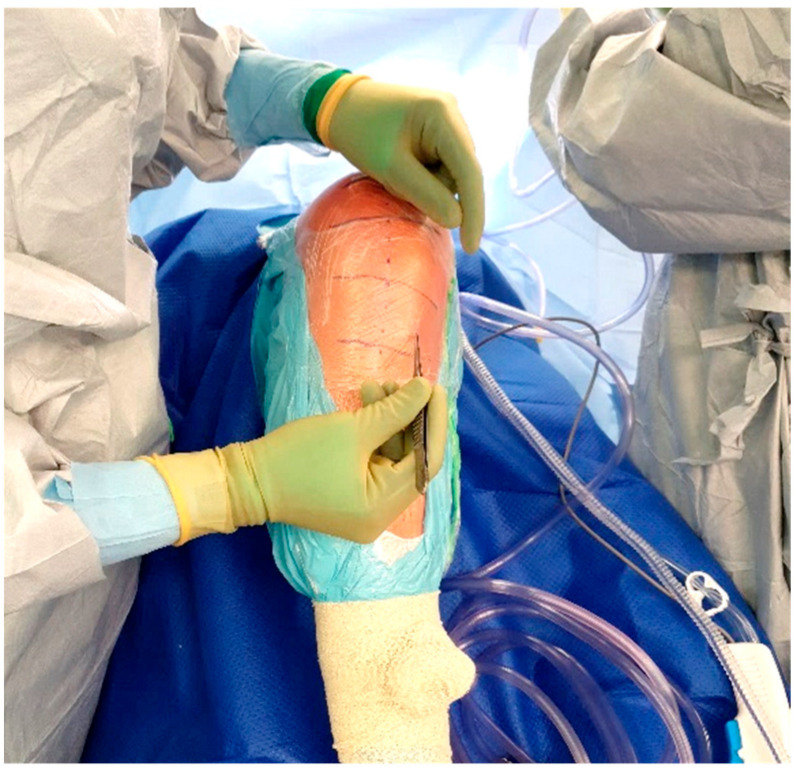
Location of IO insertion on a right knee.

**Figure 4 antibiotics-11-00634-f004:**
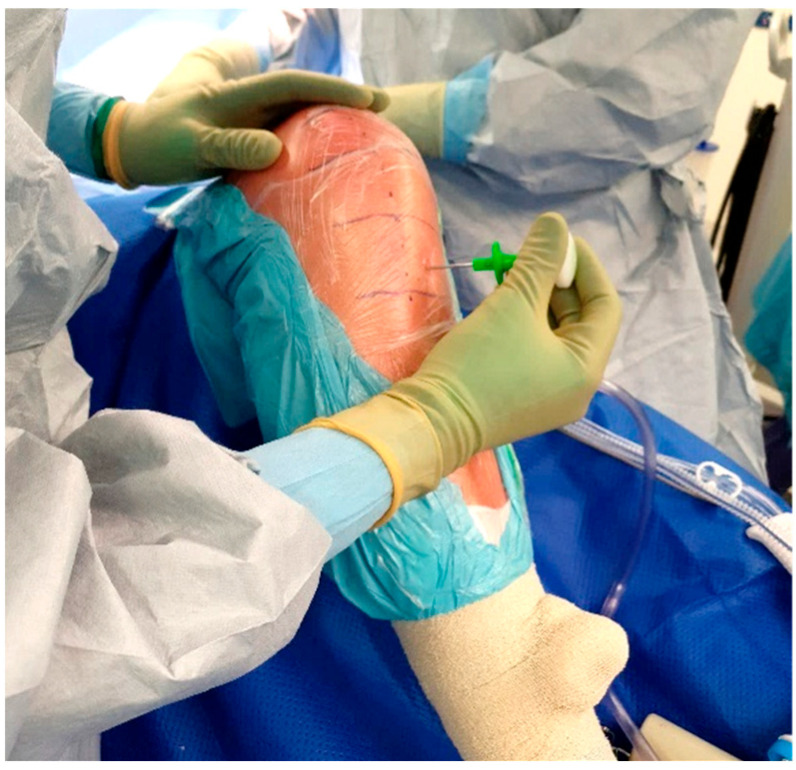
Manual IO insertion.

**Figure 5 antibiotics-11-00634-f005:**
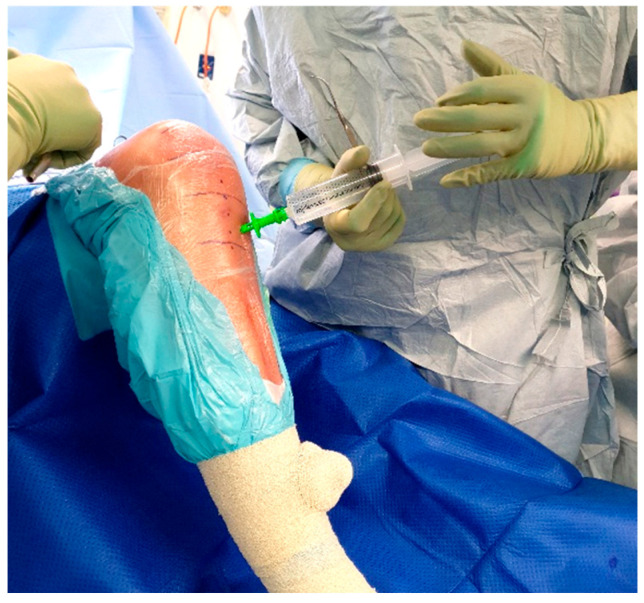
Injection of vancomycin solution.

**Figure 6 antibiotics-11-00634-f006:**
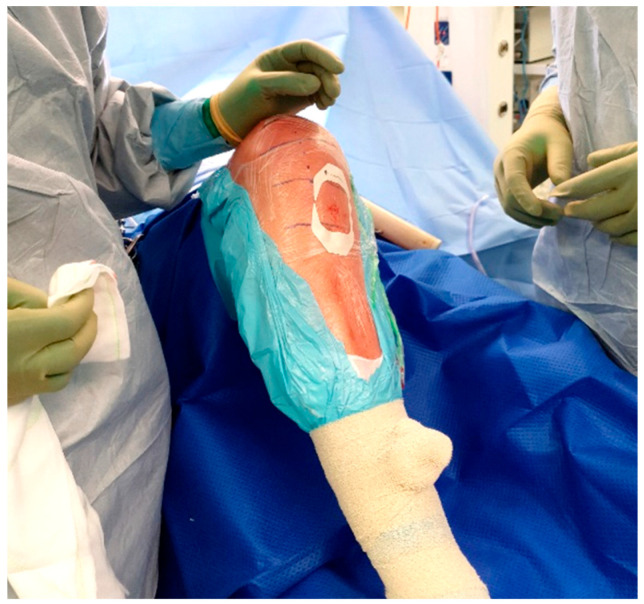
Tegaderm placement.

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
