# Peer review of "Intraosseous Regional Administration of Prophylactic Antibiotics in Total Knee Arthroplasty"

_antibiotics, 2022, doi:10.3390/antibiotics11050634_

Round 1
Reviewer 1 Report
Thank you for the opportunity to review this paper.
Few issues that need to be adressed:
The paragraph on the PK/PD should be improved.
Especially parts relating to vancomycin PK/PD - as far a I know, AUC/MIC is the index that best correlates with vancomycin efficacy
The reference is not appropriate when talking about concentration over MIC and vancomycin.
Please check, rewrite and support with adequate references.
When mentioning biofilm, a short explanation should be included.
Author Response
1. Few issues that need to be addressed:
The paragraph on the PK/PD should be improved
Especially parts relating to vancomycin PK/PD - as far a I know, AUC/MIC is the index that best correlates with vancomycin efficacy
The reference is not appropriate when talking about concentration over MIC and vancomycin.
Please check, rewrite and support with adequate references.
-------------------------------------------------
Thank you very much for your constructive feedback.
I was able to read further into this and you are correct – AUC/MIC is the index that correlates with Vancomycin efficacy – however, this is because AUC/MIC is a pharmacodynamic expression of Peak:MIC which can estimate how well concentration dependent antibiotics function. I have updated the relevant section with the addition of the following reference ; (http://www.antimicrobe.org/h04c.files/history/PK-PD%20Quint.asp) . please see lines 64 – 75. I have left the expression as peak/MIC rather than introduce the calculation of AUC/MIC. I have made it clear that this is an estimate of efficacy but I think it will make clear and simple for our readers.
2. When mentioning biofilm, a short explanation should be included
-------------------------------------------------------------------------
I thank you for picking up on this – and have included a short explanation of biofilm with appropriate reference. Please see line 332
Thank you for your feedback.
Reviewer 2 Report
In this review by Wells et al., the authors discuss the current literature related to the different methods of antibiotic administration in patients to prevent prosthetic joint infection (PJI) following total knee arthroplasty (TKA). They review the pros and cons of different delivery methods and proposed IORA to be a safe and effective choice. While scientifically, the review is sound in content, there are some mistakes and suggestions that need to be addressed.
- The review will benefit with some discussion about the late or delayed-onset PJIs and the implications of the current prophylactics including IORA in the prevention of such infections.
- Reference – Throughout the manuscript, two entirely different referencing styles have been utilized. This need to be corrected to the style accepted by the journal.
- The authors should do a thorough review for the grammar and language check for consistency and correctness.
Author Response
In this review by Wells et al., the authors discuss the current literature related to the different methods of antibiotic administration in patients to prevent prosthetic joint infection (PJI) following total knee arthroplasty (TKA). They review the pros and cons of different delivery methods and proposed IORA to be a safe and effective choice. While scientifically, the review is sound in content, there are some mistakes and suggestions that need to be addressed
- The review will benefit with some discussion about the late or delayed-onset PJIs and the implications of the current prophylactics including IORA in the prevention of such infections
------------------------------------------------------------
Thank you for your comment. As identified in our discussion, this is certainly an area that requires further investigation. This review aimed to provide an overview of current evidence on IRA use. Given that this technique is relatively new, no studies have followup longer that 12 months. Hence, we have only presented data regarding the effectiveness of IORA in preventing early (<12months) PJI.
Late onset PJIs are primarily caused by haematogenous spread, and less commonly by indolent organisms secondary to contamination at the time of surgery. Theoretically, it is possible that IVRA can reduce late PJIs by more effectively preventing infection by indolent organisms. Please refer to line 337.
2. Reference – Throughout the manuscript, two entirely different referencing styles have been utilized. This need to be corrected to the style accepted by the journal.
-----------------------------------------------------------------
Thank you for noticing this, the citation style has been updated to reflect the style accepted by the journal.
3. The authors should do a thorough review for the grammar and language check for consistency and correctness.
--------------------------------------------------------
We have reviewed the article for ease of reading and sentence structure.
Thank you for your feedback.
Reviewer 3 Report
A well-structured article. I would have liked to see more data collected and I would like to know if another technique of introducing the antibiotic at the knee has been analyzed.
In this article, only one antibiotic introduction technique and a standard dose of antibiotic were used.
In conclusion, I would like to add another antibiotic introduction technique, in order to be able to evaluate the quality of that technique, by comparison.
Author Response
A well-structured article. I would have liked to see more data collected and I would like to know if another technique of introducing the antibiotic at the knee has been analyzed.
In this article, only one antibiotic introduction technique and a standard dose of antibiotic were used.
In conclusion, I would like to add another antibiotic introduction technique, in order to be able to evaluate the quality of that technique, by comparison.
--------------------------------------------------------------
Thank you very much for your feedback. Currently, the 2 large scale retrospective trials of IORA in TKA are the only such papers that exist. Both these studies (Parkinson et. al and Park et al) use the same standardized dose and method of administration. Park et al noted that there were 20 patients (out of 488) in the early months of their study who received a lower dose of vancomycin, but they did not break down their results by this discrepancy. Similarly, there was some variation in the choice of IORA in the study by Parkinson et al, with some pateints receiving IO cephazolin, however the number were not listed. They used they technique outlined by Young et al in all cases, and the differing choice of agent was down to surgeon preference. Results were no broken down by agent, so we could not share this information in our review.
I have expanded the relevant section to include some information about different doses of antibiotic which were used in early trials. The relevant study by Young et al. showed that using a dose of 250mg Vancomycin or 500mg Vancomycin both achieved higher tissue concentrations of Vancomycin that even high dose IV prophylaxis. Please see lines 146 to 148.
Thank you for your feedback
Reviewer 4 Report
Knee arthroplasty is a widely used and highly successful procedure. One of the most serious complications is infection.
Iv antibiotic prophylaxis is the current standard of care, usually for 48h. However, several local administrations seem to provide additional benefits.
Intraosseous administration is the least known and has showed promising results that may potentially affect clinical practice.
The current clinical review comes from prestigious authors with valuable contributions on this topic.
It is written in a clinician friendly manner yet it still provides all current relevant information. Both the journal and the orthopedic community will benefit.
Author Response
Knee arthroplasty is a widely used and highly successful procedure. One of the most serious complications is infection.
Iv antibiotic prophylaxis is the current standard of care, usually for 48h. However, several local administrations seem to provide additional benefits.
Intraosseous administration is the least known and has showed promising results that may potentially affect clinical practice.
The current clinical review comes from prestigious authors with valuable contributions on this topic.
It is written in a clinician friendly manner yet it still provides all current relevant information. Both the journal and the orthopedic community will benefit.
----------------------------------------------------------------
Thank you very much for your review and your support. We appreciate you taking the time to read our paper, and are glad you have found value in it.